# Sustainable Thematic Investing: Identifying Opportunities Based on an Analysis of Stewardship Reports

**Kara Nel \*, Nadia Mans-Kemp** and **Pierre D. Erasmus**

Department of Business Management, Stellenbosch University, Stellenbosch 7600, South Africa
* Correspondence: karanel1995@gmail.com

**Abstract:** Globally, a growing number of stakeholders recognise that sustainability determines success on multiple levels. Therefore, asset managers in developing and emerging countries increasingly focus on sustainable investment opportunities. While institutional investors largely centred on governance considerations pre-2020, the Coronavirus pandemic highlighted substantial social and environmental concerns at companies worldwide. As South Africa is the most unequal country globally according to the World Bank, decisions made by local institutional investors can have significant implications for individuals and environments where capital is invested. The objectives of this study were hence to analyse the sustainability themes on which South African asset managers focused in their stewardship reports and to explore the Sustainable Development Goals (SDGs) that they addressed through their investment mandates. A content analysis was performed on stewardship reports that were published in 2020 and 2021 to consider the impact of the Coronavirus pandemic. The findings indicate that prioritised sustainability themes include climate action, infrastructure development and social considerations. The considered asset managers accordingly focused on addressing climate action (SDG 13), decent work and economic growth (SDG 8), and affordable and clean energy (SDG 7). Promising investment opportunities in companies that address key social issues, including the health and well-being of society (SDG 3) and broadening access to quality education (SDG 4) were also highlighted. The leaders of local investee companies are thus encouraged to ensure concise, transparent reporting on these material matters to enhance communication and engagement with institutional investors and other key stakeholders. This study offers a novel perspective on sustainable thematic investing in a highly unequal society.

**Keywords:** sustainability; corporate social responsibility; CSR; asset managers; institutional investors; stewardship report; investment mandate; Sustainable Development Goals

## 1. Introduction

The global market for sustainable funds has experienced exceptional growth since the advent of the Coronavirus pandemic. The value of sustainability-themed investment products in the global financial market was approximately $5.2 trillion in 2021 compared to $1.3 trillion in 2019, indicating an increase of 300 per cent over this period [1]. Environmental, social and governance (ESG) factors are integrated into the composition of sustainable funds [2,3]. Institutional investors have a particular interest in sustainable investing as they have a fiduciary responsibility to beneficiaries to incorporate all value drivers, including ESG factors, in investment decision-making [4–8]. Pertaining to this responsibility, some institutional investors also refer to the term 'responsible investing', which is used interchangeably with sustainable investing in corporate reports [9,10].

While institutional investors largely focused on governance considerations pre-2020, the Coronavirus pandemic highlighted substantial social and environmental concerns globally [11–13]. Therefore, companies started to increase their investment in corporate social responsibility (CSR) practices, thereby, inter alia, aiming to protect the livelihoods of their employees and donate food to communities in need [14,15].

There is no universal definition for CSR, as CSR is context-, country- and even stakeholder-specific. Therefore, regulatory frameworks pose challenges to the development of a universal integrated CSR framework [16]. The United Nations Principles for Responsible Investment (UN PRI) thus argued that the global business community requires a shared vision for a sustainable future, incorporating CSR. The UN introduced its 17 Sustainable Development Goals (SDGs) and 169 related targets in 2015 with the aim to provide an evidence-based framework for sustainable development planning and implementation [17]. These goals include no poverty, no hunger, good health and well-being, and climate action. The SDGs can be used as a framework to guide investments in companies, products and services that are designed to address social and environmental challenges. These goals provide a universal language for communication about sustainable development and the role of companies and investors in society [18,19].

Globally, asset managers increasingly use sustainable thematic investment strategies to identify investment opportunities that impact the environment, resource scarcity, economic inequality, and technology advances [20]. Therefore, asset managers play a vital role in the provision of resources for sustainable development since environmental and social issues are incorporated into their investment products and credit policies [21]. However, the engagement of the private sector with sustainable development has been limited because the SDGs often relate to countries and public sectors and can be difficult to apply at the individual enterprise or investor level [22]. Additionally, not all SDGs are 'equally investable' or will create substantial value for institutional investors [23] (p. 99). Researchers have found that select, financially attractive SDGs attract capital while other SDGs remain underfunded by institutional investors [24]. The Coronavirus disease of 2019 (COVID-19) especially derailed investment in the SDGs in developing countries [25]. Therefore, institutional investors who have access to substantial funds are increasingly focusing on SDG-themed investments [1]. Even though institutional investors' decisions can have a substantial impact on the individuals and environments where their capital is invested [26,27], some researchers highlighted the limited scope of investigations on strategic CSR in developing countries [20,28–30]. Therefore, the sustainability themes that asset managers in developing countries prioritise in their investment decisions warrant more attention.

South Africa is the most unequal country in the world, ranking first among 164 countries in the World Bank's global poverty database [31]. Citizens are facing numerous ESG-related challenges such as water scarcity, gender discrimination and a large pay gap [27]. There is hence increased interest from South African asset managers to incorporate ESG criteria into investment decision-making [32]. Industry bodies and the government have published several regulations and promulgated legislation to encourage sustainable investing, including amendments to Regulation 28 of the Pension Funds Act (No. 24 of 1956), the King IV Report on corporate governance and the Code for Responsible Investing in South Africa (CRISA) [4,33,34]. Nevertheless, the thematic sustainable investment opportunities that asset managers in South Africa prioritise have not yet been explored by analysing their stewardship reports.

As asset managers play a critical role in achieving the SDGs, especially in developing countries, their commitment to, engagement with, and reporting on the goals that they invest in, warrant investigation. The research objectives of this article were therefore twofold. Firstly, to identify the sustainability themes that selected asset managers who operate in South Africa focused on in their 2020 and 2021 stewardship reports. Secondly, to explore the SDGs that these asset managers addressed through their investment mandates. Identification of specific SDGs that are addressed in the considered investment mandates could highlight investment opportunities and also identify risks to help address underfunding of some of the SDGs in South Africa.

In the remainder of the article, the literature on sustainability terminology, applicable theoretical lenses, previous research, and the South African context, are reviewed. Thereafter, the content analysis that was performed on the stewardship reports is explained,

followed by a discussion of the findings on the sustainability themes and SDGs that the considered asset managers prioritised. Lastly, conclusions and recommendations for a range of stakeholders are provided.

## 2. Literature Review

Sustainability terminology, three applicable theories (communication, stakeholder, and stewardship), and an overview of CSR and sustainable development research are discussed. Hereafter, the South African asset management context is explored.

### 2.1. Overview of Key Sustainability Terminology

Given the complex nature of sustainability, it is important to understand key terms and their linkages with one another. Sustainability is commonly defined as meeting the needs of the present generation without compromising the ability of future generations to meet their needs [35]. While some scholars deem this definition ambiguous [36] others regard it as all-encompassing [37]. Corporate leaders thus often rather refer to sustainable development which describes how companies solve environmental, economic and social issues [38]. The UN (2019) SDGs provide opportunities for companies to address sustainable development and provide an integrated framework for CSR engagement [17,39]. The SDGs are intended to be universal, calling for integrative approaches that link human development and environmental sustainability by addressing issues such as poverty, inequality and environmental conservation [40]. The 17 SDGs are outlined in Table 1.

**Table 1.** The Sustainable Development Goals (adapted from [41]).

| Goal Number | Goal Name | Goal Description |
|---|---|---|
| 1 | No poverty | End poverty in all its forms everywhere. |
| 2 | Zero hunger | End hunger, achieve food security and improved nutrition and promote sustainable agriculture. |
| 3 | Good health and well-being | Ensure healthy lives and promote well-being for everyone. |
| 4 | Quality education | Ensure inclusive and equitable quality education and promote lifelong learning opportunities for everyone. |
| 5 | Gender equality | Achieve gender equality and empower all women and girls. |
| 6 | Clean water and sanitation | Ensure availability and sustainable management of water and sanitation for everyone. |
| 7 | Affordable and clean energy | Ensure access to affordable, reliable, sustainable and modern energy for everyone. |
| 8 | Decent work and economic growth | Promote sustained, inclusive and sustainable economic growth, full and productive employment and decent work for everyone. |
| 9 | Industry, innovation and infrastructure | Build resilient infrastructure, promote inclusive and sustainable industrialisation and foster innovation. |
| 10 | Reduced inequalities | Reduce inequality within and among countries. |
| 11 | Sustainable cities and communities | Make cities and human settlements inclusive, safe, resilient, and sustainable. |
| 12 | Responsible consumption and production | Ensure sustainable consumption and production patterns. |
| 13 | Climate action | Take urgent action to combat climate change and the impact thereof. |
| 14 | Life below water | Conserve and sustainably use the oceans, seas, and marine resources for sustainable development. |
| 15 | Life on land | Protect, restore, and promote sustainable use of terrestrial ecosystems, sustainably manage forests, combat desertification, and halt and reverse land degradation as well as halt biodiversity loss. |
| 16 | Peace, justice, and strong institutions | Promote peaceful and inclusive societies for sustainable development, provide access to justice for everyone and build effective, accountable, and inclusive institutions at all levels. |
| 17 | Partnerships for the goals | Strengthen the means of implementation and revitalise the Global Partnership for Sustainable Development. |

By aligning corporate goals with the SDGs outlined in Table 1, corporate leaders can, for instance, invest in renewable energy projects and education and skills development initiatives that would optimise sustainable value creation [17,42]. Sustainable development challenges can also be addressed by investing in sustainable investment funds that are designed to meet these challenges [18]. Therefore, the goals of sustainable development could be achieved through responsible investment by investing in companies that engage in strategic CSR, which specifically describes the integration of CSR practices into their core business strategies [43].

The term CSR can be broadly defined as society's economic, legal, ethical and discretionary expectations of organisations at a given point in time [44]. The first step towards being a good corporate citizen is to produce goods and services that society requires at a reasonable price [45]. Therefore, economic responsibilities arguably represent the basic building block of CSR, since a profit-making organisation is economically sustainable. Organisations should also obey the law, engage in fair and just business practices, and contribute resources to the community [45,46]. Companies can arguably 'do well by doing good' by making decisions and linking their practices to the economic, legal, ethical and philanthropic components of CSR [47] (p. 92).

Carroll's (1991) outlined definition of CSR became one of the most widely cited sources in the field of business [48,49]. However, this definition of CSR has evolved considerably since the 1990s as several stakeholders started to expect companies to engage in more than just philanthropic endeavours, and to also become involved in strategic CSR initiatives [50]. As such, the preferred definition of CSR differs given the unique contexts of industries, companies and their stakeholders [51]. The implementation of CSR policies, therefore, addresses the goals of sustainability. Investing in companies that prioritise strategic CSR offers numerous benefits to investors. For instance, a favourable CSR reputation can provide a competitive advantage [52,53]. Institutional investors can achieve better risk-return ratios through sustainable investments [5,6]. By investing responsibly, institutional investors are also adhering to their fiduciary duty towards different stakeholders and could consequently be considered societal stewards, as explained next.

### 2.2. Theoretical Lenses

The usage of multiple theoretical lenses can offer complementary perspectives on complex considerations [54]. A combination of the communication, stakeholder and stewardship theories is applicable to understanding the complexities related to sustainable investment decision-making.

Communication is the act of 'transmitting' information between individuals or organisations with the purpose to extend messages [55,56]. As social and environmental performance increasingly gain prominence, communication of CSR information is valued by investors [57,58]. A sustainability or stewardship report fosters conversations between an asset manager and its investors and other stakeholders, thereby making it a crucial communication device which stakeholders utilise in their decision-making [59]. Clear communication on their fund mandates can thus result in asset managers achieving specific economic and social goals.

The communication theory entails the study of the interaction of different elements in the communication process [56]. The communication system designed by Shannon and Weaver [56] is widely accepted and has been adopted by previous researchers who focused on financial and sustainability reporting [60,61]. The application of this communication system in the context of stewardship reporting by asset managers is outlined in Figure 1.

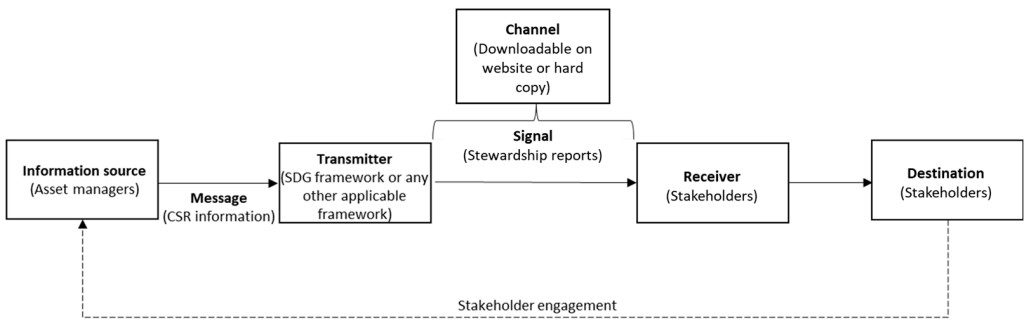

**Figure 1.** Corporate social responsibility communication process (developed for this study based on [56]).

As shown in Figure 1, asset managers (the information source) initiate the communication process and decide which CSR information will be communicated to stakeholders. The CSR information is transformed into a format such as the SDG framework which is applicable to communicate the CSR information through a stewardship report. These reports are available as downloadable portable document format (PDF) files on asset managers' websites and/or as hard copies of the reports that are sent to stakeholders.

The stakeholders are also deemed the destination in Figure 1 since they analyse and interpret the information that they receive in the stewardship reports. Different stakeholders might value divergent environmental and social impacts. Consequently, the stewardship reports offer the opportunity for stakeholders and asset managers to engage with one another to identify material impacts, thereby making the communication process interactive (as indicated by the dotted arrow in Figure 1) [62].

In the investment context, the stakeholder theory suggests that asset managers do not only prioritise their financial interests but also consider the long-term effects of creating shared value when making investment decisions [63]. Asset managers have responsibilities towards various stakeholders who are affected by their funds' mandates. Value can accordingly be created for investors by integrating stakeholders' interests into investment decisions [64].

It is evident from the discussed communication and stakeholder theories that asset managers manage relationships amongst different stakeholders since they invest on behalf of others. Therefore, asset managers also behave as stewards [65]. The stewardship theory postulates that managers do not always act in a self-interested way as suggested by the agency theory. Instead, managers might have other motivations to pursue mutual benefit [63]. The UN PRI defines stewardship as "the use of influence by institutional investors to maximise overall long-term value including the value of common economic, social and environmental assets, on which returns and clients' and beneficiaries' interests depend" [66]. Investors can therefore act as 'stewards of society' by encouraging companies to balance the interests of various stakeholders [67] (p. 993).

### 2.3. Overview of Corporate Social Responsibility and Sustainable Development Research

Several researchers investigated the CSR practices that are most valued by investors in developed countries. In Japan in 2013, selected individuals, financial institutions and brokerages were more concerned with social contributions than environmental protection [68]. Similarly, certain responsible investment mutual funds in the United States held significantly higher ownership in firms that focus on employee relations and social CSR aspects than environmental and governance considerations in 2017 [69]. In 2020, selected asset managers in the United States deemed corporate political activity and equal employment key investment concerns [70].

In contrast, institutional investors in South Korea perceived environmental practices as more important than social considerations when making investment decisions post-2020 [29]. They focused on investing in companies that address pollution and waste, and greenhouse gas emissions. Although German investors tended to focus on companies

that alleviate poverty and create good working conditions in a survey conducted in 2013, individual respondents differed on which factors should be excluded from sustainable investment funds [71].

With reference to SDGs, the United Nations Global Sustainability Index Institute (UNGSII) foundation found that responsible consumption and production (SDG 12) and climate action (SDG 13) were prioritised by the 500 largest investments made by companies globally in 2019. The UNGSII noted that poverty reduction (linked to SDG 1) and conservation of oceans (associated with SDG 14) were underfunded [72]. Refer to Table 1 for a detailed description of the SDGs.

All SDGs are arguably not 'equally investable' in terms of the available investment opportunities [23]. Schramade [23] rated the SDGs according to global investment opportunities and the results showed that SDGs 3, 7, 8, 9 and 12 presented the most investment opportunities, while SDGs 1, 2, 4, 13, 14 and 15 either produced no investment opportunities without other enabling SDGs being present or produced very few investment opportunities for asset managers by 2016. Likewise, a global study conducted amongst pension funds [73] showed that health care (linked to SDG 3) and technology (linked to SDG 9) present more investment opportunities than natural resources (associated with SDG 15). Based on a literature review of policy and development documents and interviews with sector experts, the United Nations Development Programme (UNDP) reported that agriculture (linked to SDG 2), health care (related to SDG 3), quality education (SDG 4) and infrastructure (SDG 9) offered the most sustainable investment opportunities in South Africa [74].

A study on the disclosure of SDGs by selected Spanish financial institutions in their sustainability reports during the period 2016–2019 found that they prioritised investments in two SDGs, namely quality education (SDG 4) and decent work and economic growth (SDG 8) as both goals contribute to job creation and combatting unemployment [75]. Likewise, PricewaterhouseCoopers (PwC) indicated that the financial sector globally prioritises SDG 8 [76].

Although an international investment community group comprising 60 high-net-worth private investors that are managed from the United States, and Germany preferred to invest in SDGs that are associated with high financial returns, they disagreed on which SDGs would result in the highest financial returns. The majority of these investors considered SDG 14 (life below water), SDG 15 (life on land) and SDG 5 (gender equality) as the least investable. Therefore, these SDGs remain underfunded. The researchers suggested that other impact-oriented investor groups that have economic significance to fund sustainable development, such as asset managers, warrant more investigation [26].

Limited research has been conducted on strategic CSR and sustainable thematic investing in developing countries [29,30], especially on the African continent. Scholars also noted that there is a lack of CSR studies covering institutional investors [77], specifically studies that investigate their CSR disclosure [78]. More research is needed on the sustainability reporting of asset managers worldwide to understand the contributions made by the financial services sector to sustainable development (via the SDGs) and the impact on society, the environment and the economy [26,76]. For the purpose of this study, the focus was placed on South Africa, as explained next.

*2.4. South African Context*

There is growing participation by South African institutional investors to apply ESG criteria to investment decision-making [33]. The country has various regulations and legislation in place to guide such practices, including amended Regulation 28 of the Pension Funds Act (No. 24 of 1956), the King IV Report and CRISA [4,34,35]. South African pension funds are legally required to account for ESG aspects that could materially affect investments [79]. Regulation 28 sets limits to particular asset classes to protect retirement fund members against investment risk. The latest amendment of Regulation 28, which has been effective since January 2023, enables retirement funds to increase their investments in

international assets. The investment limits between hedge funds and private equity have also been split to further facilitate long-term infrastructure and economic development [80].

The King IV Report also focuses on sustainable value creation which encourages companies to account for an ethical culture, good performance, effective control and legitimacy. Principle 17 of this report encourages the governing bodies of institutional investor organisations to ensure that responsible investment is practiced as outlined by CRISA [33]. In 2011, CRISA was launched to encourage South African institutional investors to integrate ESG factors in their investment decisions. Accordingly, local institutional investors were urged to invest with a long-term perspective that incorporates social and environmental factors. In 2022, CRISA was updated to include five principles, namely the integration of ESG factors, diligent stewardship, capacity building and collaboration, sound governance and transparency [81].

Decisions made by institutional investors can hold significant social and environmental implications for a range of stakeholders [26,27]. Therefore, the stewardship reports of selected South African asset managers were investigated in this study to identify the sustainability themes that they focused on in their investment decisions as well as to explore the SDGs that these asset managers addressed through their investment mandates.

## 3. Materials and Methods

Qualitative research is typically conducted to determine different stakeholders' preferences and evaluation of specific corporate practices [82]. The stewardship practices and responsibilities of asset managers are disclosed in their annual stewardship, responsible investment, impact or sustainability reports [83]. These annual reports usually explain the different investment funds that they offer and include a section on their approach to sustainable investing. Secondary qualitative data were therefore collected, and a content analysis was conducted on the 2020 and 2021 stewardship reports to investigate the sustainability themes that selected local asset managers prioritise. The sample selection, data analysis, and trustworthiness considerations, are discussed in the following sections.

### 3.1. Sample Selection and Data Analysis

Asset managers invest in companies on behalf of beneficiaries and have a fiduciary responsibility to make responsible decisions regarding investment analysis, activities and returns [4]. In contrast, individual investors purchase shares in their personal capacity [84]. The asset managers in the sample all had institutional investors as clients. Institutional investors include pension funds, insurance companies and hedge funds. Institutional investors have more opportunities to address sustainability concerns than individual investors, as they manage large funds. Additionally, asset managers can considerably influence companies' policies and practices by engaging with corporate leaders and using their voting power [1].

By August 2022, 98 discretionary financial service providers were registered with the South African Reserve Bank (SARB) according to its institutional investor list [85]. However, only 12 of these asset managers provided access to their 2020 and 2021 stewardship reports on their websites and could hence be included in the sample. The research period (2020–2021) was selected since the Coronavirus pandemic emerged in South Africa in March 2020. This pandemic highlighted key environmental and social challenges and therefore more focus was placed on sustainability concerns since 2020 than in previous decades [13–15]. The COVID-19 crisis also created the opportunity to identify investment opportunities to achieve the SDGs.

Prior scholars have conducted content analyses on sustainability reports to analyse CSR activities in other settings than South Africa pre-2020 [86–88]. Some researchers have also classified selected companies' contributions to SDGs by conducting a content analysis as it provides a systematic approach to describe or identify patterns in communication in line with the discussed communication theory [89–91].

The sustainability themes on which the sampled asset managers focused were compared for the year in which the Coronavirus pandemic emerged and in 2021. The stewardship reports were downloaded in PDF format to conduct a content analysis in ATLAS.ti 22.

The stewardship reports were firstly reviewed to determine whether the selected asset managers were UN PRI signatories and subscribed to CRISA for purposes of descriptive discussion in the findings section. The asset managers that were not UN PRI signatories and that did not subscribe to CRISA were not excluded from the sample. Secondly, the reports were read in full to identify the sustainability themes and SDGs that were discussed in the reports. Thirdly, the number of sustainability-focused funds were counted per report.

Lastly, deductive coding was conducted based on predefined key words and phrases to address the two research objectives. The reports and included fund mandates were coded according to an iterative process whereby the identified categories were refined after each round of coding. Several rounds of coding were conducted, as some asset managers explicitly reported the SDGs that their funds addressed, while others used descriptions and project examples. After each round of coding, the decision criteria were refined to determine which sustainability concern(s), theme(s) and SDG(s) were addressed by a specific fund based on the analysed text.

Pertaining to the first research objective, key words and phrases were used to identify the prioritised CSR activities to ultimately determine which sustainability themes selected asset managers that operate in South Africa focused on in their stewardship reports published during the COVID-19 pandemic. The utilised key words and phrases were based on a CSR practice list provided by the CSRHub [92]. This list includes thirteen CSR practices, namely volunteering in the community, respecting human rights, providing equal compensation for work of equal value and standing against discrimination regarding race, gender and/or religion, providing a healthy work environment, addressing climate change, reducing carbon footprint, producing safe products, reporting on environmental impact, acting in an environmentally friendly manner, having a diverse board of directors, having healthy relationships with stakeholders and not engaging in unethical practices such as bribery and corruption. The CSR data schema was deemed appropriate as funds that address a certain social or environmental concern address the core purpose of strategic CSR. The CSRHub's CSR practice list was deemed suitable since it is applicable to South African companies. Hereafter, the investment mandates were coded to address the second research objective based on the SDG framework outlined in Table 1.

During the deductive coding data collection phase, it was observed that more than half of the considered asset managers explicitly reported on the SDGs that they have addressed in their fund mandates. Moreover, the asset managers that did not explicitly report on the SDGs, used similar wording than the SDG framework to describe the social and environmental aspects that they addressed in their fund mandates. The few asset managers who have not explicitly reported on the SDGs or used the SDG wording in their fund mandates, mentioned the social and environmental concerns that were included in the list of CSR activities from the CSRHub [92]. The discussed CSR activities were then linked to applicable SDGs. There were only three CSR activities that did not match the exact SDG wording. The CSR practice of equal pay were hence linked to gender equality (SDG 5). The practice of providing a healthy work environment was equated to decent work and economic growth (SDG 8). Reduction of carbon footprint was linked to climate action (SDG 13). Each of these practices form part of the targets that are addressed in the related goals and therefore it was deemed appropriate to link them to the respective SDGs.

Each CSR practice or SDG was only coded once in a report. Therefore, the number of times that each SDG was mentioned was not considered but rather whether the SDG was mentioned in the respective reports.

### 3.2. Trustworthiness and Ethical Considerations

Lincoln and Guba's [93] criteria for trustworthiness, which include credibility, dependability, confirmability, and transferability were applied. Pertaining to credibility, a

recognised, applicable research method, namely a content analysis was used to analyse the qualitative secondary data. To enhance dependability, a detailed description of the research design and methodology has been provided in Section 3 to ensure that the study could be repeated in future in other contexts. The coding decisions were explained in Section 3.1 to enhance confirmability. The transferability of the findings might be limited in the global context, as this study focused on the investment mandates of asset managers with publicly available stewardship reports in South Africa during 2020 and 2021. Quotes from the stewardship reports were included in Section 4, where applicable, to support the derived findings.

Ethical clearance was obtained before collecting the data. Pseudonyms were assigned to the asset managers in the following findings section to protect their anonymity.

## 4. Discussion of the Findings

The findings of the content analysis cover the sustainability themes that selected asset managers, who operate in South Africa, focused on during 2020 and 2021 (as disclosed in their stewardship reports, in line with the first research objective). Hereafter, the SDGs that the considered asset managers addressed through their investment mandates are explained.

### 4.1. Sustainability Themes in Stewardship Reports

The UN PRI encourages asset managers globally to make responsible investments and enhance risk management to create sustainable value over the long-term [94]. Likewise, CRISA urges local asset managers to integrate ESG issues in their investment decisions. Table 2 provides a summary of the sampled asset managers who were UN PRI signatories and supported CRISA during the period under review. A summary is also provided of the respective names of their annual reports.

**Table 2.** Summary of adherence to regulatory frameworks and reporting statistics (developed by the authors based on the content analysis).

| Year | UN PRI Signatory | Support CRISA | Stewardship Report | Responsible Investment Report | Impact Report | Sustainability Report |
|------|------------------|---------------|--------------------|-------------------------------|---------------|-----------------------|
| 2020 | 100% | 83.33% | 58.33% | 8.33% | 16.67% | 16.67% |
| 2021 | 100% | 83.33% | 58.33% | 16.67% | 16.67% | 8.33% |

All the considered asset managers were UN PRI signatories in 2020 and 2021, whereas ten of the 12 (83.33%) sampled asset managers indicated in their analysed reports that they supported CRISA. Although the other two asset managers indicated on their websites that they support CRISA, they did not explicitly mention this information in their analysed reports. In line with the discussed communication theory [56], the analysed signal (stewardship reports) should foster communication between the communication source (asset managers, in the case of this study) and the receivers of the communication, namely stakeholders. It is thus important to include relevant information in annual reports and not just on websites.

The terminology used to refer to the reports published on an annual basis by the considered asset managers were not consistent. Although the majority of the sampled asset managers (58.33%) published a stewardship report, the others referred to a responsible investment, impact or sustainability report, as shown in Table 2. In South Africa, there is not yet a standard practice regarding the preferred name or structure of such a report.

Even though all the asset managers in the sample were signatories of the UN PRI and the majority adhered to CRISA, the layout and disclosure of sustainability concerns were not standardised. Substantial differences in the so-called transmitter and signal could challenge communication, and by implication engagements on key sustainability matters between asset managers and divergent stakeholders in the local context, in line with the discussed communication theory [56].

The global social and environmental concerns that were highlighted by the COVID-19 pandemic [11,12] were reflected in the South African context based on the analysed stewardship reports. For example, asset manager B indicated in their 2020 stewardship report that the "COVID-19 crisis has revealed the need to place more emphasis on social sustainability as a key part of the investment outlook." Likewise, asset manager I stated that their strategy in 2021 was to invest capital with the intention of not only generating financial returns but also having a social and an environmental impact. These statements of the considered asset managers are in line with the outlined stakeholder theory [64] which postulates that asset managers should not only consider financial returns, but also prioritise the long-term effects of their investment decision-making on society and the environment.

Asset manager D furthermore explained in their 2020 stewardship report that the COVID-19 pandemic elevated the conversation on social inequality in South Africa. As such, this asset manager provided details on engagements with investee companies covering transformation considerations. It was evident that several local asset managers had to reconsider their decision-making practices when the pandemic arose, as illustrated by the following quote from asset manager B's 2020 stewardship report:

> "We look forward to embracing the significant changes brought on by 2020 and continuing to make a relevant and meaningful impact across all our investment activities".

For some of the considered asset managers, these changes meant increased focus on sustainable development. For example, asset manager C observed in their 2020 stewardship report that investors increasingly focus on assessing their investment portfolios based on the extent to which it is contributing to the achievement of the SDGs. The content analysis furthermore showed that some of the considered funds explicitly focused on social and environmental concerns. Asset manager E stated in their 2021 report that "investments made via these funds have all had a positive social and environmental impact including improved infrastructure, job creation, environmental benefits, housing, access to basic services, water and improved health care". Several of the considered asset managers furthermore encouraged investee companies to balance the interests of their divergent stakeholders. In line with the discussed stewardship theory [67], they hence seemingly acted as responsible stewards of the assets under their management.

Table 3 indicates the number of funds that were managed by the sampled asset managers that explicitly focused on environmental and social concerns. These funds covered climate action, infrastructure, and social considerations in 2020 and 2021 and are thus referred to as sustainability-focused funds.

**Table 3.** Themes covered by sustainability-focused funds (developed by the authors based on the content analysis).

| Year | Theme | | | Total Funds ‡ |
|------|-------|---|---|---------------|
| | Climate Action | Infrastructure | Social Considerations † | |
| 2020 | 2 | 5 | 7 | 14 |
| 2021 | 4 | 10 | 9 | 23 |

† Social considerations include job creation, education, health care and inequality. ‡ The funds were managed by five asset managers in 2020 and six asset managers in 2021.

As shown in Table 3, the total number of sustainability-focused funds increased with approximately 64 per cent from 14 funds in 2020 linked to five asset managers to 23 funds across six asset managers in 2021. More funds thus explicitly focused on social and environmental matters a year after the advent of the COVID-19 pandemic in comparison to 2020. A possible reason for this trend is the increased demand for social and environmental-focused investment products globally [12].

The investigated asset managers also increased the variety of their sustainability-focused funds over the research period. One of the asset managers had six impact funds

that covered different social and environmental concerns in 2021, compared to only one sustainability-focused fund in 2020.

Furthermore, the funds that focused on climate action doubled from 2020 to 2021 (refer to Table 3). Yet the climate theme still had the lowest number of funds that exclusively focused on this prominent sustainability concern by 2021. A possible reason for this trend could be that it is challenging to construct a fund that purely focuses on climate issues in a country where there are limited low-carbon investment opportunities [95]. The funds that centred on infrastructure noticeably increased over the research period, thereby indicating that infrastructure was a particularly important sustainability theme for institutional investors since the advent of the COVID-19 pandemic.

The funds that focused on social issues also increased with about 29 per cent, as shown in Table 3. The first research objective, namely to identify the sustainability themes that selected asset managers that operate in South Africa focused on in their 2020 and 2021 stewardship reports was addressed in this section. Details on how the sampled asset managers addressed the discussed SDGs are provided next.

### 4.2. Sustainable Development Goals Addressed in Investment Mandates

Almost 60 per cent of the considered asset managers reported explicitly on the SDGs that their investment funds addressed in 2021, compared to only 50 per cent in 2020. This increase indicates that more of the considered asset managers directly linked their social and environmental investment mandates to the SDGs post-2020. This finding offers support for the notion that the SDGs provide an acceptable framework for sustainability reporting to simplify communication with investors, as suggested by [19] and [40].

In line with the themes identified in Section 4.1, the 16 funds linked to social considerations for the entire research period addressed, on average, five SDGs in their fund mandates. The maximum number of SDGs that a considered social impact fund contributed to was 14 of the 17 goals. In comparison, the portfolios in a global study contributed, on average, to six SDGs. The maximum number of SDGs captured in a portfolio in the global study was 15 [24]. The SDGs were also clustered to address a specific social problem, which in turn dealt with the core purpose of strategic CSR.

As mentioned in Section 3.1, the considered investment mandates were coded according to the SDG framework outlined in Table 1. Figure 2 indicates how many of the sampled asset managers focused on each SDG in their investment fund mandates in 2020 and 2021, respectively. The relevant percentages are also indicated.

Worland (2020) [96] postulated that action related to climate change might decrease in the future, as institutions globally are largely focused on economic recovery since the advent of COVID-19 pandemic. It is, however, evident from Figure 2 that the majority of the sampled asset managers centred on climate action (SDG 13) in their investment fund mandates in 2020 (83.3%) and 2021 (66.7%). This finding of the current study is in line with the outcomes of studies by [29] (conducted in South Korea) and [73] (covering a global sample). Despite the noticeable decline over the research period for SDG 13, the substantial annual percentages shown in Figure 2 for this goal can be linked to the pledge that South Africa made in 2020 to achieve net zero carbon emissions by 2050 [97]. Since then, several local asset managers joined the voluntary agenda for climate change [98–100].

In line with their focus on climate change, most of the sampled asset managers disclosed that they have followed the Task Force on Climate-Related Financial Disclosures (TCFD) recommendations during the period under review. Some of the asset managers even produced a separate TCFD report. The TCFD was established to develop recommendations for the types of climate-related information that companies should disclose to support investors to make more informed investment decisions [101]. Net zero initiatives were also mentioned in some of the analysed reports.

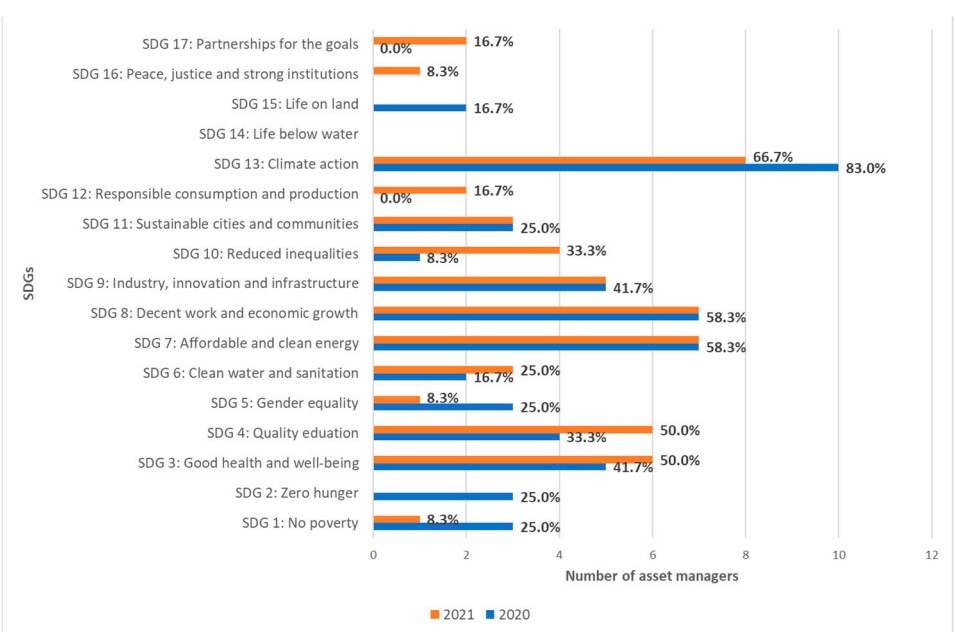

**Figure 2.** Sustainable development goals mentioned in investment mandates.

All the considered asset managers indicated that they did not follow an exclusion strategy. This strategy entails excluding companies from an investment portfolio based on a range of social and environmental criteria [102]. However, most of the sampled asset managers indicated that they have engaged with representatives of specific investee companies to address environmental issues. Many of the considered asset managers indicated that they have used the TCFD recommendations when they have engaged on such issues. This finding relates to the increased number of funds that focused on climate action concerns (see Table 3). The increase in the inclusion of clean water and sanitation (SDG 6) in some of the considered investment mandates between 2020 and 2021 is encouraging as South Africa is a water-scarce country. The UNDP reported that the local government indicated that SDG 6 is amongst the SDGs that they prioritise [74].

Furthermore, the considered asset managers contributed to decent working environments and economic growth (SDG 8) as well as invested in affordable and clean energy (SDG 7) in 2020 and 2021. These findings also correspond with the SDGs that the South African government prioritise [74]. Globally, the financial services sector also mainly concentrates on SDG 8 [76]. This trend can be largely ascribed to the significant role played by financial institutions in promoting investments in projects that create jobs and combatting unemployment [75]. For example, asset manager I substantially contributed to job creation in South Africa by assisting an investee company to open 14 new stores in only four months in 2021, with the prospects of opening another 25 stores with 75 related work opportunities within a year of their initial investment. Similarly, asset manager J invested R18 million towards job creation in 2020, thereby offering access to economic opportunities for more than 18,000 individuals.

There are furthermore several investment opportunities in the public and private sectors for renewable energy provision (linked to SDG 7) in South Africa. The country had to embrace alternative energy resources given the severe energy crisis. Some of the asset managers in the sample regarded the renewable energy sector as more favourable than other sectors from an ESG perspective.

For instance, asset manager G explained in their 2020 stewardship report that they have identified SDG 7 as an attractive investment avenue, particularly solar and wind projects in South Africa that meet both their return and impact investing objectives. Asset manager B likewise commented in their 2020 report that "renewable energy projects provide good returns for our investors". Their portfolio consisted of more than 20 per cent of all renewable energy projects in South Africa in 2021, which included 17 power

plants that generated, on average, 3,200,000 megawatt-hours of renewable energy per annum. Likewise, asset manager I invested R145 billion in solar projects in 2020, thereby contributing clean renewable energy to power more than 19,000 South African households. Furthermore, asset manager K not only invested in 11 wind and solar projects over the period under review, but they also financed a R225 million renewable energy plant in a remote area of South Africa as part of a project to accelerate growth following the outbreak of COVID-19. The UN Conference on Trade and Development's World Finance Report of 2022 [1] confirmed that investments in renewable energy projects were responsible for the overall growth both in the number of projects and in the value of SDG-related investments in developing countries. Asset managers are liable to their clients to make the most responsible decisions relating to financial returns [4–8]. Paetzold, Busch, Utz and Kellers [24] concurred that asset managers prefer to invest in SDGs that results in good financial returns.

As shown in Figure 2, gender equality (SDG 5) received more attention in 2020 than in 2021. Despite the increase in 2021, very limited attention was given to reduced inequalities (SDG 10) in 2020. This result is concerning given that South Africa is the most unequal country globally with prominent inequality issues [31,103]. In contrast to the findings of the current study, PwC indicated that the financial services sector ranked SDG 5 (gender equality) as the fifth most prioritised goal, globally [76]. Although the percentage of women on the boards of companies and key government positions have increased worldwide, women are still underrepresented in these positions. Furthermore, while venture capital and equity allocated to women founders increased globally, the allocated amounts remain low [104]. Asset manager I was one of the few asset managers in the sample that prioritised SDG 10. They indicated in their 2021 report that they have continued to combat inequality by providing "equity type finance to medium-sized South African companies in support of business growth and to ensure jobs are retained in the short term and increased in the longer term".

Other key social issues in 2021 included good societal health and well-being (SDG 3) and broadening access to quality education (SDG 4) (refer to Figure 2). These findings are in line with the view of the UNDP (2020) that SDGs 3 and 4 offer some of the most promising sustainable investment opportunities in South Africa [74]. Financial institutions in Spain likewise concentrated on quality education [75]. Investing in education is important for economic growth as it leads to social upliftment by alleviating poverty and creating job opportunities. Asset manager B explained that their investment mandate addressed quality education in 2020 by investing R3.1 million in education initiatives through investee companies' CSR and socio-economic development projects. They stated in their 2020 stewardship report that:

> "These projects improve early childhood development, provide equal access for all women and men to affordable, good-quality technical and vocational education and training (TVET), including tertiary education at universities, and increase the number of people with relevant qualifications for decent employment and entrepreneurship".

Asset manager H similarly indicated in their 2020 stewardship report that they invested R2.3 billion in projects that provided access to quality education to more than 18,000 children across South Africa. Globally, financial services institutions ranked quality education (SDG 4) third and health care (related to SDG 3) fourth out of the 17 goals in terms of priority [76].

More than 40 per cent of the considered asset managers invested in infrastructure (SDG 9) in 2020 and 2021. Asset manager B stated in their 2021 stewardship report that such an investment is "tangible and makes a sustainable difference to South African communities". One of their investee companies had construction and rehabilitation contracts worth R1.25 billion that contributed to infrastructure development in 2020. Infrastructure investments allow asset managers to capitalise on opportunities presented by government's infrastructure plans. Additionally, infrastructure-related investments are encouraged by

amended Regulation 28 that seeks to direct increased pension fund capital to this goal [80]. Investments in specific SDGs can increase based on government regulation and policy [73].

Pertaining to environmental-related SDGs that require attention, none of the considered asset managers focused on SDG 14 (life below water) in their investment mandates during the entire period under review, whereas only 16.7 per cent of them invested in SDG 15 (life on land) in 2020. This finding is consistent with the observations of other researchers that investments in natural resources are at the lower end of the investment spectrum [24,72,73].

Similarly, from a social perspective, none of the considered asset managers prioritised SDG 2 (zero hunger) in 2021. Institutional investors seem uncertain about how to address SDG 2 in practice since the extent of investment opportunities in this regard that can be accessed through listed companies are limited [23]. Likewise, the UNGII [72] found that SDG 2 has extremely limited visibility globally in annual reports. Nevertheless, food security and hunger are major concerns in South Africa as a quarter of the South African population are living below the food poverty line. Furthermore, severe stunting (related to low height-for-age values) of 30 per cent was reported for children under the age of five in South Africa in 2022. The implication is that three out of every 10 children in the country are suffering from chronic undernutrition [105]. As none of the considered investment funds directly addressed SDGs 2, 14 or 15 in 2021, these goals remain underfunded in South Africa.

Furthermore, none of the sampled asset managers prioritised SDG 16 (peace, justice, and strong institutions) in 2020. The same applies to SDG 17 (partnerships for the goals). In contrast, SDG 16 was one of the top sustainability concerns that asset managers in the United States prioritised in 2020 [70]. Despite limited attention being given to SDG 16 in this study, South Africa received a below average score of 44 (where 0 is indicating a highly corrupt and 100 is indicating a 'very clean' country) on the 2021 Corruption Perception Index [106]. It is thus evident that SDG 16 warrants more attention from local asset managers.

## 5. Conclusions and Recommendations

The stakeholder and stewardship theories show that the investment decisions that asset managers make have a substantial impact on pension fund members, society at large and the environment. Even though prior researchers have conducted studies on the linkages between and the implications of strategic CSR and corporate sustainability in developed countries, the contributions made by institutional investors to sustainable development through making sustainable thematic investments in developing and emerging countries warrant investigation.

South African asset managers increasingly incorporate ESG criteria in their investment decision-making given growing regulatory and stakeholder pressure to account for ESG considerations in this emerging market. Yet South Africa is the most unequal country around the globe according to the World Bank [31]. South African citizens are facing numerous sustainability challenges such as water scarcity, gender discrimination and a large pay gap. The commitment of selected South African asset managers to sustainable development was hence investigated in the context of the COVID-19 pandemic by conducting content analysis on their publicly available 2020 and 2021 stewardship reports.

Key findings were reported based on the content analysis. Most of the asset managers were UN PRI signatories and subscribed to CRISA. It was evident that the sampled asset managers placed considerable emphasis on sustainable development since the advent of the COVID-19 pandemic, as more sustainability-focused funds were developed in 2021 compared to 2020. In terms of the sustainability themes on which the considered asset managers focused in their analysed stewardship reports (the first research objective), the findings indicate that the main themes include climate action, social considerations, and infrastructure development.

To address the second research objective, the SDGs that the sampled asset managers addressed through their investment mandates were identified. The findings show that the considered asset managers largely focused on climate action (SDG 13), affordable and clean energy (SDG 7), as well as decent work and economic growth (SDG 8) when making investment decisions. Additionally, the considered asset managers expected investee companies to address key social issues, including societal health and well-being (SDG 3) and broadening access to quality education (SDG 4). The sampled asset managers hence seemed to have adopted a focused investment approach during the period under review to make progress in meeting specific SDGs. They concentrated on goals that they deemed critical in addressing the most prominent sustainability challenges in South Africa.

The findings can encourage corporate leaders of companies operating in South Africa to focus on sustainability practices related to the SDGs that institutional investors value and deem critical to enhance sustainable development in the country. Corporate leaders can accordingly attract funding from institutional investors for projects related to reducing their carbon emissions and renewable energy provision. Additionally, corporate projects related to enhancing access to health care and quality primary, secondary and tertiary education in South Africa are likely to attract attention from institutional investors. These considerations should thus be pertinently mentioned in companies' CSR policies and reports.

The UN PRI, and other applicable professional associations can account for the reported findings when evaluating the South African investment community's progress in addressing the SDGs. Reflection on the SDGs that were highlighted in the considered asset managers' investment mandates could assist representatives of such bodies to identify promising investment opportunities in the local investment context. Additionally, reasons for the evident underfunding of specific SDGs by selected asset managers in South Africa, including SDGs 2, 14 and 15 could be explored by industry bodies.

The South African government, the King Committee (that updates the King Report) and the Association for Savings and Investment South Africa (that develops CRISA) are encouraged to increasingly engage with institutional investors regarding future amendments to legislation and industry guidelines to further enhance sustainable investment and development. Likewise, corporate leaders are urged to ensure concise, transparent reporting on material matters to enhance communication and engagement with institutional investors and other key stakeholders on sustainability concerns.

This study has two main limitations. Firstly, only secondary qualitative data were analysed by conducting content analysis. In the future, researchers could conduct interviews with selected asset managers to discuss their views on prominent sustainability themes and CSR practices on which they base their investment decisions in selected emerging markets. Reasons for the underfunding of SDGs could also be discussed. Future scholars could furthermore explore why asset managers focus on specific sustainability themes and SDGs by accounting for personal preferences, education, and attitude of individual asset managers by using a survey. Quantitative assessments of SDGs could also be explored, including the development of measuring instruments by incorporating suitable proxy indicators. The application thereof can then be tested by conducting case studies in developed and developing countries, Secondly, this study focused on a limited number of South African asset managers that made their stewardship reports publicly available on their websites. Future researchers can explore the sustainability-focused practices of a larger sample of asset managers operating in various emerging markets, such as South Africa, Brazil and China by utilising a survey.

Based on the communication theory, the application of the UN SDG framework to analyse stewardship reports were illustrated in the South African emerging market context. Future researchers are encouraged to make use of this framework and theory when they account for corporate and asset management reporting on sustainability matters in other countries. This study contributes to the emerging body of research on sustainable development by showing which SDGs play a prominent role in the sustainable thematic

investment decision-making of an economically significant stakeholder group, namely asset managers in the most unequal country in the world.

**Author Contributions:** K.N. conceptualised the article and prepared the draft manuscript, including the methodology and data analysis; K.N., N.M.-K. and P.D.E. edited and finalised the manuscript; supervision: N.M.-K. and P.D.E.; project administration: K.N. All authors have read and agreed to the published version of the manuscript.

**Funding:** K.N was supported by a doctoral scholarship from Stellenbosch University's Graduate School of Economic and Management Sciences (GEM). The funder had no role in study design, data collection and analysis, decision to publish, or preparation of the manuscript.

**Institutional Review Board Statement:** The study was approved by the Research Ethics Committee of Stellenbosch University (ONB-2022-24473).

**Informed Consent Statement:** Not applicable.

**Data Availability Statement:** Data sharing is not applicable as no new data were created in this study.

**Acknowledgments:** We sincerely thank the four anonymous reviewers for their valuable inputs.

**Conflicts of Interest:** The authors declare no conflict of interest.

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
