# Peer review of "Sustainable Thematic Investing: Identifying Opportunities Based on an Analysis of Stewardship Reports"

_sustainability, doi:10.3390/su15108411_

Round 1
Reviewer 1 Report
Recommendation:. Major Revision
Comments
Thank you for giving me the opportunity to review the manuscript titled, “Sustainability Themes That Asset Managers Prioritise: An Analysis of Stewardship Reports” Below you can find my comments in detail. My hope is that my comments be useful for your endeavor
1) Your title seems disconnected regarding the asset management, if I analyze it correctly, I am not sure how a sustainable investment opportunities factors are linked. Perhaps you should link them.
2) There are few things about the abstract which need to be reevaluated. Abstract should be more informative presenting clear aim as well as study novelty. Study motivation should be more focused on ‘asset management and sustainable investment opportunities in developing countries. Moreover, the line mentioned “The findings indicate that South African asset managers focus on climate change, decent working conditions, economic growth and affordable,clean energy when making investment decisions.”. This is not clear as of now its complicate.
3) In introduction section the motivation behind the study is not clear. There is 10 paragraphs, reduce the paragraph and link to them. For example paragraph 4 & 6 not clear. Moreover, the line mentioned “Companies that invested in CSR were more resilient during the Coronavirus pandemic”. This is not clear as of now companies are based on the different types.
4) Literature review – The author tried any recent data or not? For example, line mentioned “By aligning corporate goals with the UN SDGs, corporate leaders can invest in projects that would optimise sustainable value creation [19,51].” it may be unclear which types of projects ? How these changes will be incorporated into this paper.
5) Material and Method section is not sufficiently explained, without any linkage paragraph explained [a major issue]. Explain in detail how author conducted test and trials.
6) Results: The authors provided the detailed results of the study. It will be pleasing if results are linked to existing literature which would help in strengthening the conclusions.
Overall, I do think the paper might need to improve the consistency to maintain the quality of the journal.
All the best!

Reviewer 2 Report
The topic of the manuscript is relevant, and its content may be of interest to potential readers. The authors' detailed review of literary sources should also be included among the positive features of this manuscript. However, there are lot of gaps and shortcomings that authors need to remove before a manuscript can be published:
1. Main remarks:
1.1. The authors of the article do not sufficiently fully and clearly describe the scientific novelty of the results they obtained. How do these results differ from the results obtained earlier by other scientists?
1.2. The empirical results presented by the authors are mainly descriptive in nature. The authors do not explain the reasons for the differences between different asset managers in their attitude towards certain sustainable development goals. What are the reasons for these differences (perhaps personal preferences, education, motivation, etc.)?
1.3. Also, the authors mainly consider the declared goals, rather than the specific actions of asset managers. Have the tasks been implemented? In particular, it would be interesting to have information about the number of relevant projects, the amount of investment, the share of these amounts in the total amount of invested investments, etc.
1.4. The size of the sample raises doubts about the representativeness of the results obtained by the authors.
2. Other remarks:
2.1. In my opinion, it is not necessary to start subsection 2.1 with a description of the essence of corporate social responsibility. It is better to first provide a description of the essence of sustainable development, its features, criteria, etc.
2.2. The names of some sections and subsections should be adjusted. In particular, the name of section 4 is not very good. It is better to call it "Results and their discussion".
2.3. It is worth describing the deductive coding procedure in more detail (line 306).
2.4. Sections 3 and 4 should be fully consistent with each other. However, I do not see in section 4 the results of applying the criteria for trustworthiness that the authors mention in subsection 3.2.
2.5. Section 4 provides a number of quotations. It is not always possible to understand exactly who the authors are quoting and how much these quotes characterize the entire sample.
2.6. It is unclear the size of the sample, the data for which are shown in Table 2. How many funds were considered?
2.7. The text of the manuscript sometimes lacks proper connections between paragraphs. Presentation of all material should be consistent. For example, in the conclusions before line 555, it would be appropriate to write something like this: "Our research made it possible to obtain a number of results."
2.8. Style and grammar need some improvement. In some parts of the text, the style is not completely scientific. In particular, I don't recommend starting your introduction with a quote. Also, the wording of some sentences could be improved. For example, the wording of such a sentence (lines 126-128) is not very successful: “Corporate leaders thus rather refer to sustainable development, as sustainable development can be used to describe how companies solve environmental, economic and social issues [46]”.
I think it is appropriate to acquaint the authors with these comments, suggestions and questions. I hope that such acquaintance help to improve the quality of the manuscript, which is expected to be published in such a high-ranking journal as "Sustainability".
Reviewer 3 Report
Dear Author(s),
This is a timely effort to examine the ESG practice amongst SA asset managers in relation to their investment clients. The paper is well-structured and written with clarity. The only weakness of this project is the justification of timing for data collection, i.e. why has the research window of 2020-2021 been selected? Is it the consideration of the impact of Covid-19? If so the title needs chaning by specifying the impact of Covid-19. Otherwise you need to expand the period by a few years back as the SDGs were prublished in 2015. A better strategy might be that you provided a time series in a table to show the starting year of the ESG investment emerged in SA. Whichever the way you need to improve the justification of data collection.
Good luck.
Reviewer 4 Report
I recommend the paper with minor revision. Here are some comments:
1) What are some of the key sustainability themes that asset managers prioritize, according to the analysis of stewardship reports?
2) How do institutional investors differ from other types of investors in their approach to sustainable investment opportunities?
3) Are there any notable differences in the sustainability priorities of asset managers in developed versus developing countries?
Round 2
Reviewer 2 Report
In my opinion, the text of the manuscript has improved. The authors took into account most of my comments. At the same time, the text of the manuscript still has certain shortcomings (mostly of a technical nature) and some debatable points, namely:
1. Line 16 states that "As South Africa is most unequal country". The same statement is given several more times in the text of the article (in particular, in lines 81, 632). Isn't this statement too categorical?
2. I ask you to check very carefully the correctness of the use of all abbreviations in the text of the manuscript. In particular, this applies to the acronym ESG. It applies only to types of factors. Therefore, is it correct to use this abbreviation, for example, in the following phrase in line 276: “to integrate ESG in their investment decisions”? It is also necessary to check whether all abbreviations are deciphered at the first mention, for example PwC. I believe that all abbreviations, even well-known ones, should be deciphered (explained).
3. Also, please check the correct use of the word "They" in some sentences (lines 25, 64, etc.). In some cases, it is difficult to immediately understand who is being talked about.
4. It is necessary to check the correctness of the wording of some sentences. In particular, this applies to sentences in lines 79-80, 164-166 (the statement is debatable), 246-247, 281-285 (I recommend splitting this sentence into two sentences, and also indicating that it refers to this particular study, and not to previous researches of other scientists), 288-290, 340-343, 399-402, 619-621.
5. In lines 110 and 111, we are talking about sustainability or about sustainable development?
6. You need to edit the text in lines 148-150 (extra lines?).
7. It is not entirely clear which code is being referred to in line 278. The phrases “in the first term” in line 312 and “to decent work environments” (to decent working conditions?) in lines 519-520 are also not entirely clear to me.8. Can some of the funds, the data on which are presented in the table. 2, cover several themes at the same time? From the material in line 517 it follows that this is impossible.
9. I propose to detail the title of Figure 2. Where exactly are these goals indicated?
10. If in section 4 the authors mention the results of research in other countries (for example, in Spain in lines 570-571), it is necessary to indicate whether these results are consistent with those obtained by the authors of this article, or contradict them.
11. In the conclusions, the authors sometimes use the phrase "Future researchers (scholars)". It is better to use the phrase "In the future researchers (scholars)".
Reviewer 3 Report
Dear Authors,
Your revision through further work following the recommendations is satisfactory.
